METHODS

# Inferring gene regulatory network from single-cell transcriptomes with graph autoencoder model

Jiacheng Wang[1,2], Yaojia Chen[1], Quan Zou[1,2]*

1 Institute of Fundamental and Frontier Sciences, University of Electronic Science and Technology of China, Chengdu, China, 2 Yangtze Delta Region Institute (Quzhou), University of Electronic Science and Technology of China, Quzhou, Zhejiang, China

* zouquan@nclab.net

**Data Availability Statement:** Data availability All data underlying the findings are fully available without restriction and can be accessed by the following URLs: The in silico scRNA-seq dataset, six synthetic networks and four curated networks

## Abstract

The gene regulatory structure of cells involves not only the regulatory relationship between two genes, but also the cooperative associations of multiple genes. However, most gene regulatory network inference methods for single cell only focus on and infer the regulatory relationships of pairs of genes, ignoring the global regulatory structure which is crucial to identify the regulations in the complex biological systems. Here, we proposed a graph-based Deep learning model for Regulatory networks Inference among Genes (DeepRIG) from single-cell RNA-seq data. To learn the global regulatory structure, DeepRIG builds a prior regulatory graph by transforming the gene expression of data into the co-expression mode. Then it utilizes a graph autoencoder model to embed the global regulatory information contained in the graph into gene latent embeddings and to reconstruct the gene regulatory network. Extensive benchmarking results demonstrate that DeepRIG can accurately reconstruct the gene regulatory networks and outperform existing methods on multiple simulated networks and real-cell regulatory networks. Additionally, we applied DeepRIG to the samples of human peripheral blood mononuclear cells and triple-negative breast cancer, and presented that DeepRIG can provide accurate cell-type-specific gene regulatory networks inference and identify novel regulators of progression and inhibition.

## Author summary

Although many methods have been proposed to infer the gene regulatory network of a single cell, they only focus on the regulatory relationships of pairs of genes and ignore the global regulatory structure. Here, we present a deep learning-based model to learn the global regulatory structure and reconstruct the gene regulatory networks from single-cell RNA sequencing data with a graph view. We utilize the weighted gene co-expression analysis to build a prior regulatory graph of gene and a graph autoencoder to deconstruct the latent regulatory structure among genes. We performed extensive experiments on varieties of single-cell RNA sequencing datasets and compared our method with 9 stat-of-the-art gene regulatory network inference method. The results show that our method can

analyzed during current study can be accessed by BEELINE (https://doi.org/10.5281/zenodo.3378975). The gene expression matrix of scRNA-seq datasets analyzed in this study were downloaded from Gene Expression Omnibus with the accession numbers GSE75748 for hESC dataset, GSE98664 for mESC dataset, GSE48968 for mDC dataset, GSE81682 for mHSC dataset. The ground truth for six real scRNA-seq datasets derived from ChIP-seq data can be derived from ChIP-Atlas (https://chip-atlas.org/) and be accessed by BEELINE (https://doi.org/10.5281/zenodo.3378975). The scRNA-seq dataset for PBMC8k was downloaded from the website for 10x genomics (https://www.10xgenomics.com/resources/datasets/8-k-pbm-cs-from-a-healthy-donor-2-standard-2-1-0). The scRNA-seq datasets for the normal breast cells and TNBCs were downloaded from Gene Expression Omnibus with the accession number GSE118390. The ground truth regulatory networks of PBMCs and normal breast cells were downloaded from hTFtarget database (http://bioinfo.life.hust.edu.cn/hTFtarget#!/download). Code availability DeepRIG is open source and publicly available at GitHub (https://github.com/JChander/DeepRIG).

**Funding:** This work is supported by the National Natural Science Foundation of China (62131004 and 62250028 to QZ; 62301369 to JW), and a Municipal Government Found of Quzhou (2022D040 to QZ). The funders had no role in study design, data collection and analysis, decision to publish, or preparation of the manuscript.

**Competing interests:** The authors have declared that no competing interests exist.

significantly improve the accuracy of gene regulatory network inference and can be applied to identify key regulators in a wide range of scenarios.

## Introduction

Compared to bulk RNA-sequencing (RNA-seq) technology, single-cell RNA-seq (scRNA-seq) data offers the opportunity to identify the cell heterogeneity and the biological process of dynamic cellular differentiation [1]. Within scRNA-seq data, the reconstruction of gene regulation networks (GRNs) can successfully uncover the cellular differentiation mechanisms [2]. A GRN is the regulatory interaction structure that depicts the complex biological systems in the level of genes. It is formed by the interactions between genes and genes within a cell or within a particular genome, especially those based on gene regulation. Unfortunately, inferring GRNs from single-cell transcriptomics is challenging, since the gene expressions of single cells are highly variable.

To address this issue, dozens of computational algorithms have been developed to infer GRNs from scRNA-seq data. As a consequential observation, the strategy that constructing the correlation-based co-expression networks of TFs and their target genes has been widely adopted to infer the GRNs. Many of these methods are based on correlation coefficients [3, 4] or mutual information [5, 6] to measure the co-expression modes of genes. Qiu et al. integrates RNA dynamics modeling, robust vector field reconstruction, and geometric analyses to construct and interpret single-cell transcriptomic vector fields, enabling insights into cell-fate transitions and genetic perturbations [7]. Since pseudotime and time-stamped single-cell transcriptional profiles are considered to provide the dynamic information that reflects cellular differentiation, some methods [8–11] apply the regression models to measure the changes of gene expression using pseudotime-order data or the time-stamped data, and several other methods [12, 13] employ linear ordinary differentiation equation (ODEs) to model the regulatory dynamics. Schiebinger introduced Waddington-OT, an innovative approach that uses developmental time courses from single-cell RNA sequencing to infer ancestor-descendant fates, model regulatory programs, and gain insights into diverse developmental processes, including reprogramming [14]. Furthermore, methods including SCENIC [15], IRIS3[16], and CeRIS [17] have been developed to identify the active regulons of specific cell types and to further reveal underlying GRNs in diverse diseases. Recently, deep learning models have also been proposed to capture the complex non-linear regulations from scRNA-seq data by using convolutional neural network (CNN) [18–20] and variational autoencoder [21]. While these methods have made remarkable progress, however, most of them only focus on the regulatory relationships of pairs of genes and fail to leverage the intrinsic global regulatory structure of a GRN that is crucial to explore the regulatory modes between TFs and their target genes in the complex biological systems.

We introduce DeepRIG to reconstruct the GRNs from single-cell transcriptomics by using a graph autoencoder (GAE) model. To learn the global regulatory structure, DeepRIG takes the trick of calculating the correlation coefficients for each pair of genes from gene expression profiles of scRNA-seq data, and builds a prior regulatory graph based on the co-expression mode which is robust to noise. Converting the GRNs into gene graphs is an effective way to take advantage of its intrinsic global structure, which can decouple the complex regulatory relationships among genes by integrating their neighborhood information. Nodes in the graph denote genes, and edges denote their regulatory relationships. And we further transformed the GRNs inference task into the prediction of missing edges in the graph. Meanwhile, we utilize

the GAE to embed the global regulatory information into the gene latent representations and to reconstruct the GRNs by incorporating the prior regulatory graph and expression profiles of genes. DeepRIG can accurately infer the GRNs in an end-to-end manner with a semi-supervised way, only depending on gene expression data and a small labeled set of positive pairs (often called the ground truth).

We extensively evaluated the performance of DeepRIG for GRN inference on both *in silico* datasets generated by BEELINE [22] and scRNA-seq datasets from real cells. Benchmarking results demonstrated DeepRIG can reconstruct the GRNs accurately and outperforms 9 state-of-the-art GRN inference methods. We further validated that the GAE model can greatly improve the GRN inferring performance by comparing to a straightforward WGCNA model. In addition, we used DeepRIG to infer the cell-type specific GRNs from human peripheral blood mononuclear cells (PBMCs), and identified several hub TFs and marker genes of CD14 + monocytes and B cell, suggesting that DeepRIG can be assisted in determining cell type dependent cell-type specific GRNs. We also applied DeepRIG to the samples of triple-negative breast cancer (TNBCs) and observed ten genes with significant topological characteristics by analyzing the inferred GRN. Four of them have been reported to be involved in the development, progression and inhibition of TNBC, which demonstrates that DeepRIG enables the discovery of novel regulators or targets in the complex biological systems of disease.

## Description of the method

### Inferring gene regulatory networks with DeepRIG

As an end-to-end framework, DeepRIG takes the gene expression profiles of single-cell data as input, and generates a regulatory score matrix by explicitly modeling the regulatory network structure between highly variable TFs and candidate genes using GAE model (Fig 1). DeepRIG started to preprocess the gene expression profiles of single-cell RNA-seq datasets and filter those genes that expressed in insufficient cells and "low-quality" cells. Next, DeepRIG

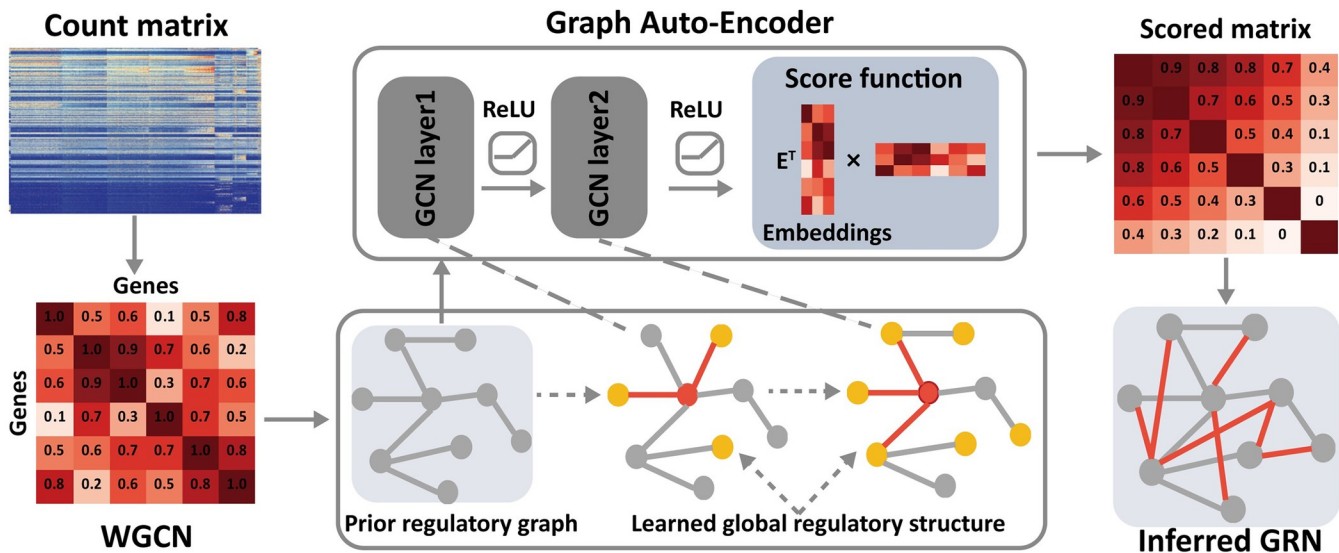

**Fig 1. The workflow of DeepRIG for inferring the gene regulatory networks.** Given a scRNA-seq dataset, DeepRIG starts to construct a weighted gene co-expression network (WGCN) from gene expression data. A gene prior regulatory graph is built according to the WGCN. The prior regulatory graph and the node feature matrix (gene expression profiles) are then inputted to a graph autoencoder (GAE). The GAE model consists of graph convolutional networks (GCNs) and a scoring function. GCNs act as encoder that learns and embeds the global regulatory structure into the latent representations of genes, and a scoring function acts as a decoder that scores each pair of learned representations of genes to reconstruct the GRNs.

extracted the co-expression modes of genes from preprocessed gene expression datasets. At this step, we calculated the Spearman's rank correlation coefficients between each gene pair among every single-cell to construct a weighted gene co-expression network (WGCN). Using the WGCN as prior knowledge, we built a prior regulatory graph of genes. Driven by this prior regulatory graph, DeepRIG utilizes a GAE model consisting of a graph convolutional network (GCN) structure and a scoring function, to effectively learn the global regulatory structure and reconstruct the GRNs from the gene expression profiles. Comparing to original auto-encoder, GAE model can further incorporate functional information provided by the prior regulatory graph in addition to gene expression data. At this step, a two-layer GCNs structure that we applied can directly decode the global graph structure into gene embeddings and discover the non-linear regulation which behind the linear WGCN (see "Methods" section). While the GCNs transformed the weighted co-expression modes of all genes to the interpretable hidden representations, the scoring function acted as a decoder and generated a regulatory score matrix of genes based on the hidden representations. DeepRIG further ranked the TF-gene pair from the matrix according to their regulatory score and reconstructed the regulatory network of genes.

## Datasets and preprocessing

We evaluated the performance of DeepRIG on GRN inference task by following the standards of BEELINE [22]. In the framework of BEELINE, both *in silico* datasets and real scRNA-seq datasets were employed to evaluate the GRN inference performance. *In silico* datasets contain single-cell gene expression datasets from six synthetic networks and four curated networks. In this paper, we also included both *in silico* datasets and real scRNA-seq datasets for the performance evaluation.

### *In silico* datasets from synthetic networks

Although the synthetic regulatory networks serving as ground truth may not reflect the complex regulatory mechanism in real cell developmental process, it is still the only realistic ground truth that contains complete information about regulatory relationships. To this end, we employed six synthetic networks served as the ground truth networks. Each synthetic network corresponds to a kind of trajectory commonly observed during cell differentiation and development, including Linear (LI), Linear Long (LL), Cycle (CY), Bifurcating (BF), Bifurcating Converging (BFC), and Trifurcating (TF), containing 7 genes, 18 genes, 6 genes, 8 genes, 10 genes, and 8 genes, respectively. The network densities of the six synthetic networks are 0.17, 0.06, 0.2, 0.24, 0.17, and 0.3, respectively. BEELINE generated 30 different *in silico* single-cell gene expression datasets for each synthetic network with varied number of cells: 10 repeats for 500 cells, 10 repeats for 2000 cells, and 10 repeats for 5000 cells by using BoolODE [22]. We used these 30 in silico single cell gene expression datasets to evaluate the performance of DeepRIG on synthetic networks.

### *In silico* datasets from curated networks

While the synthetic networks provide the complete regulatory information, however, they cannot mimic the regulatory relationships of real cells. To this end, we further employed four curated networks serving as ground truth that contain gene regulatory relationships in four developmental models, namely, mammalian Cortical Area Development (mCAD)[23], Ventral Spinal Cord Development (VSC)[24], Hematopoietic Stem Cell Differentiation (HSC) [25], and Gonadal Sex Development (GSD)[26]. For each curated model, BEELINE used BoolODE [22] to generated 10 different in silico datasets containing 2000 cells. In order to closely

reflect the characteristics of single-cell transcriptomic datasets, three rates of dropout ($q = 0$, $q = 50$, $q = 70$) were introduced into each in silico dataset.

In summaries, four curated network models were considered in the performance evaluation. We applied 30 in silico datasets with 2000 cells generated by BEELINE for each curated model, where 10 without dropout, 10 with dropout rate of 50, and 10 with dropout rate of 70.

## Real single cell RNA-seq datasets

In this paper, we further employed six real scRNA-seq datasets containing one cell line from human and five cell lines from mouse: hESC (human embryonic stem cells)[27], mDC (mouse dendritic cells)[28], mESC (mouse embryonic stem cells)[29], and three lineages from mouse hematopoietic stem cells mHSC-E (erythroid lineages), mHSC-L (lymphoid lineages) and mHSC-GM (granulocyte-macrophage lineages)[30]. We also gave a summary of the above six scRNA-seq datasets for detailed information (S1 Table).

For each real scRNA-seq dataset, we applied the gene regulatory network derived from chromatin immunoprecipitation sequencing (ChIP-seq) data of ChIP-Atlas [31] serving as the ground truth for evaluating the performance on gene regulation inference.

For six real scRNA-seq datasets, we performed the preprocessing applying Scanpy [32] pipeline. We only kept the high-quality cells and those genes which expressed in more than 90% cells. The count library sizes were normalized through transcripts per million reads (TPM) and count per million reads (CPM), and further transformed to log(TPM + 1) and log (CPM +1). Then we followed the standards of BEELINE and scanned genes for each dataset by using Bonferroni correction with P value of 0.01. We further selected TF genes from RegNetwork [33] and TRRUST [34] databases. All TFs were kept under consideration in the analysis for each dataset. In results, TFs and top 500 highly variable non-TF genes specific for each cell type were selected for the GRN inferring task after removing duplicate genes.

## Building prior regulatory graph by weighted gene co-expression network analysis

Gene co-expression networks building from the assessment of expression similarities can capture the simultaneously high or low expression modes between genes, which provides crucial clues in uncovering regulatory factors for lineage programming [35]. For the gene co-expression networks construction, nodes denote genes and edges connect two genes when they are significantly co-expressed across multiple samples. A challenge of using binary metrics to construct the gene co-expression network is how significantly the gene co-expression can be defined as a connection. Weighted gene co-expression network (WGCN) [36] is a classic and common strategy to address this issue. In WGCN framework, soft thresholding is applied to weight the connection strength of each gene pair. Comparing to binary assigned gene co-expression network, WGCN can make use of co-expression patterns and provide more important information for predicting the significant regulatory associations between genes.

In this way, the highly correlated gene pairs expressed in multiple biological samples are usually measured and connected by a variety of correlation coefficients. Spearman's rank correlation is one of the most popular correlation coefficients that quantifies the monotonic possibility between two continuous variables, rather than a strictly linear relationship. In a monotonic relationship, two variables tend to change at the same time, but not necessarily at a constant rate. Due to such desired property, Spearman's rank correlation is often considered to exhibit more robustness and tolerance to technical noise [37]. This character makes Spearman' correlation more suitable for scRNA-seq datasets, which are extremely sparse or noised due to the technical limitations during sequencing and quantifying.

In DeepRIG, we built a prior regulatory graph of genes by constructing the WGCN based on Spearman's correlation coefficients (SPCC) from the modeled expression data of single-cells. More specifically, given two log-transformed (log (TPM + 1) or log (CPM + 1), where TPM or CPM represents transcripts or counts per million) expression counts of genes $i$ and $j$, $X_i$ and $X_j$, their Spearman's correlation score is calculated as follows:

$$r_{i,j} = \frac{cov(R_{X_i}, R_{X_j})}{\sigma_{R_{X_i}} \sigma_{R_{X_j}}} = \frac{\sum_{n=1}^{N}(R_{Xi,n} - \bar{R}_{X_i})(R_{Xj,n} - \bar{R}_{X_j})}{\sqrt{\sum_{n=1}^{N}(R_{Xi,n} - \bar{R}_{X_i})^2} \sqrt{\sum_{n=1}^{N}(R_{Xj,n} - \bar{R}_{X_j})^2}} \quad (\text{AUTONUM} *\text{Arabic})$$

Where $n = \{1,2,\ldots,N\}$ denotes the number of single-cells and $R_{X_i}$ is the rank of the gene $i$ expression in cells. Notably, $r_{i,j} \in [-1,1]$, where 1 indicates completely positive correlation between gene $i$ and gene $j$, -1 indicates completely negative correlation, and 0 indicates correlated independence. The Spearman's correlation coefficients for all gene pairs are calculated and measured as the edges between genes of co-expression network. The selection of a soft threshold is crucial in WGCNA. However, there is no consensus on the optimal soft threshold for all tasks, as it depends on the specific situation and analysis objective. In this study, we chose to retain all the correlation coefficients instead of adopting other predefined strategies. This approach was motivated by the expectation that it would capture a broader range of potential regulatory information (S4 Fig). Since the weights of edges in the gene graph are non-negative, we defined correlation edges based on the absolute value of correlation coefficients. Thus, we could treat the WGCN as a prior functional graph, where the genes were denoted to nodes and the absolute value of correlation coefficient between two genes was denoted to the edge weight between two nodes in the graph. We thought that the prior regulatory graph contains the global regulatory structure about the functional similarities that provides crucial clues in uncovering regulatory factors for lineage programming.

## Graph auto-encoder reconstructs gene regulatory structure

Driven by the prior regulatory graph, we used the graph auto-encoder (GAE)[38] model to learn and embed the global regulatory structure into the gene latent representations from gene expression profiles and reconstruct the regulatory network. The adjacency matrix $A \in \mathbb{R}^{M \times M}$ of the prior regulatory graph is inputted into the GAE model, as well as the gene expression profiles. In contrast to original auto-encoder (such as DeepSEM [21]), GAE model also incorporates additional functional information provided by prior regulatory graph that is usually not presented in the gene expression profiles. GAE model combines the graph convolutional neural network (GCN) with the autoencoder, using GCN as the encoder.

GCN model [39] is a powerful neural network architecture that has been widely used in link prediction from graph. A GCN layer generalizes the convolution operations from traditional image data to graph data. By learning a function map, it can aggregate the features of a node in the graph and all the features of its neighborhood nodes in graph to generate its hidden representation. Then it uses a non-linear activation function to transform the outputs. By stacking multiple GCN layers, we can aggregate the node features from not only its direct neighborhood, but also its indirect neighborhood, which means its multiple-hop neighbors. This operation enables genes receiving messages from multiple-hop neighbor genes in the prior regulatory graph, so that we could extract the global regulatory structure and non-linear regulatory information among genes. For example, giving two regulations denoting as "x→y, y→z", our goal is to identify the non-linear regulation "x→z". By applying a two-layer GCNs, the 1-th layer can aggregate information from its direct neighbor y for x, and the 2-th layer can

aggregate information from its indirect neighbor z for x with y as the hop. Thus, a GCN with just two layers can identify non-linear regulation from x to z.

We follow the linear formulation of GCNs from the work of Kipf and Welling [39]:

$$H^{l+1} = ReLU(\widetilde{D}^{-0.5} \widetilde{A} \widetilde{D}^{-0.5} H^l W^l) \quad \text{(AUTONUM *Arabic)}$$

where $\widetilde{A} = A + I_M$ is the adjacency matrix with added self-connections, and $I_M \in \mathbb{R}^{M \times M}$ denotes the identity matrix of gene relational graph. $\widetilde{D}_{ii} = \sum_j^M \widetilde{A}_{ij}$ is the entry $i$ of degree matrix, denoting the degree of gene $i$. $W^l$ is a trainable weight matrix specific to each layer. $H^l \in \mathbb{R}^{M \times c_l}$ is the convolved signal matrix of the $l^{th}$ layer and the feature matrix inputted into the $(l+1)^{th}$ layer, and $c_l$ denotes the dimension of the feature vector for every gene. In DeepRIG, we set gene expression profiles as the node features $H^0$. $ReLU$ denotes Rectified Linear Unit, a non-linear activation function. The linear formulation of GCNs is a layer-wise approximation of spectral graph convolutions. To address overfitting on local neighborhood areas of graph, GCNs share the parameters of the graph Laplacian spectrum over the whole graph with the normalized expression $I_M + D^{-0.5} A D^{-0.5}$. However, this layer-wise operation in deep neural network may lead to the problem of gradient explosion. To address this issue, GCNs employ a trick called "renormalization", transforming above expression to $\widetilde{D}^{-0.5} \widetilde{A} \widetilde{D}^{-0.5}$. Then the features of genes connected by edges in the graph are weighted and summed to update the signal feature vector of each gene. These filtering operations are efficiently implemented as matrix multiplications.

After obtaining the convolved signal representations of genes $e \in \mathbb{R}^c$ (c denotes the dimension of the representations), a score function acts as a decoder for producing a score for each pair of genes and constructing the regulatory network structure. In DeepRIG, we follow the work of GAE and employ a single DistMult factorization [40] to score each pair of genes. For gene $i$ and gene $j$, the single DistMult function is formulated as:

$$score(i,j) = e_i^T W e_j \quad \text{(AUTONUM *Arabic)}$$

where $e_i^T$ denotes the transpose of the convolved representation $e_i$, and $W \in \mathbb{R}^{c \times c}$ is a $c$-dimensional trainable weighted matrix. Similarly, the scoring operations for all pairs of genes can be efficiently implemented as $HWH^T$.

For prediction of gene regulatory associations, the optimized loss function can be formulated as follows:

$$\mathcal{L} = -\frac{1}{|Y|} \sum_{(i,j,y) \in Y} y \, log \, l(score(i,j)) + (1-y) log(1 - l(score(i,j))) \quad \text{(AUTONUM *Arabic)}$$

Where $Y$ is the total set of samples that have labels, $l$ denotes the logistic sigmoid activation, and $y$ is the label of the samples. 1 indicates positive sample while 0 indicates negative one.

## Implementation of DeepRIG

In DeepRIG, we input the gene prior regulatory graph and gene expression profiles into the GAE model. The key hyperparameters including the number of neurons of each layer and the number of epochs for training are determined through the grid search. As a result, a three-layer GAE model was constructed, of which each layer composes 200 neurons. And we train the models for 500 epochs and use an early stopping strategy to avoid overfitting that the training procedure will be stopped if there is no improvement for the validation loss in 10 consecutive epochs. For hyperparameters optimization of each dataset, we apply the Adam optimizer

[41] with a learning rate of 0.01. The weight matrices in the models are initialized at random followed the work of Glorot and Bengio [42]. In addition, we apply L2 regularization with a penalty of 1e-4 for the first GCN and apply dropout with a rate of 0.7 for all the GCNs to further avoid overfitting problem. All hyper-parameters are summarized in S2 Table.

## Train and test dataset split

For each dataset, we split the ground truth to generate the training and test sets with random by performing 3-fold cross validation. Note that there is no intersection between training set and test set for preventing information leakage. As a link prediction task, inferring GRNs is generally class-imbalanced that the number of negative samples (unobserved edges) in the network is much larger than the number of positive samples (observed edges). For this issue, we trained DeepRIG model with negative sampling strategy. In specifically, the gene pairs with ground truth (connected) were denoted as the positive samples. Then the gene pairs without ground truth (unconnected) were randomly sampled and generated as negative samples with the same number of positive samples.

With the trained model of DeepRIG, the regulatory network covering all genes is predicted and indicated by a regulatory score matrix. Each element in the score matrix is an inferred regulatory association between two genes. We ranked the regulatory association between genes in the score matrix by using the absolute value of elements as the probability of being a potential TF-gene regulation. And all predicted regulatory relationships in the scored matrix were considered in the evaluation. Note that this whole procedure was repeated for 10 runs with random initializations.

## Evaluation metrics and baselines

We applied four numeric measures used in BEELINE to quantitative evaluate the performance on GRN inference of DeepRIG. In addition to two common metrics of the area under the receiver-operating characteristic curve (AUROC) and the precision-recall curve (AUPRC), early precision ratio (EPR) and AUPRC ratio values defined by BEELINE [22] are also computed to evaluate the inferring ability of true positives and AUPRC values by considering the network dense. The EPR and AUPR-R denote the ratio of the true positives and AUPRC values in top k inferred edges for that model and a random predictor, where k represents the number of edges in the ground truth network.

We benchmarked DeepRIG against the 9 state-of-the-art GRN inferring methods, PIDC [5], PPCOR [4], LEAP [3], GRNBOOST2[9], SCODE [13], SCRIBE [6], DeepSEM [21], CNNC [18], 3DCEMA[20]. Note that another deep learning-based GRN inferring method CNNC [18] is a supervised method and outputs only a fraction of all TF-gene pairs on testing. For a fairer comparison on evaluation, we adapted its outputs to a ranked edge list of all TF-gene pairs and kept the genes contained in the output of all competing methods consistent. Similar to DeepRIG, the weights of potential edges inferred by 9 competitive methods are considered as confidence of regulatory associations and used to rank the lists of edges. As Several GRNs inference methods, including PIDC, GRNBOOST2, and DeepSEM, output unsigned regulatory networks, making it challenging to differentiate between activating and inhibitory regulations. Moreover, the ground truth ChIP-seq data also provides unsigned regulatory associations, further complicating the evaluation. To address this limitation, we focused on evaluating unsigned GRNs of three types of ground truth for all baseline methods. In contrast, our model provides a significant advantage by outputting signed regulatory networks, representing both activating and inhibitory regulations, which closely mirror the complexity of real biological networks. A positive score in the predicted interaction indicates an activating regulation, while a negative score signifies an inhibitory one.

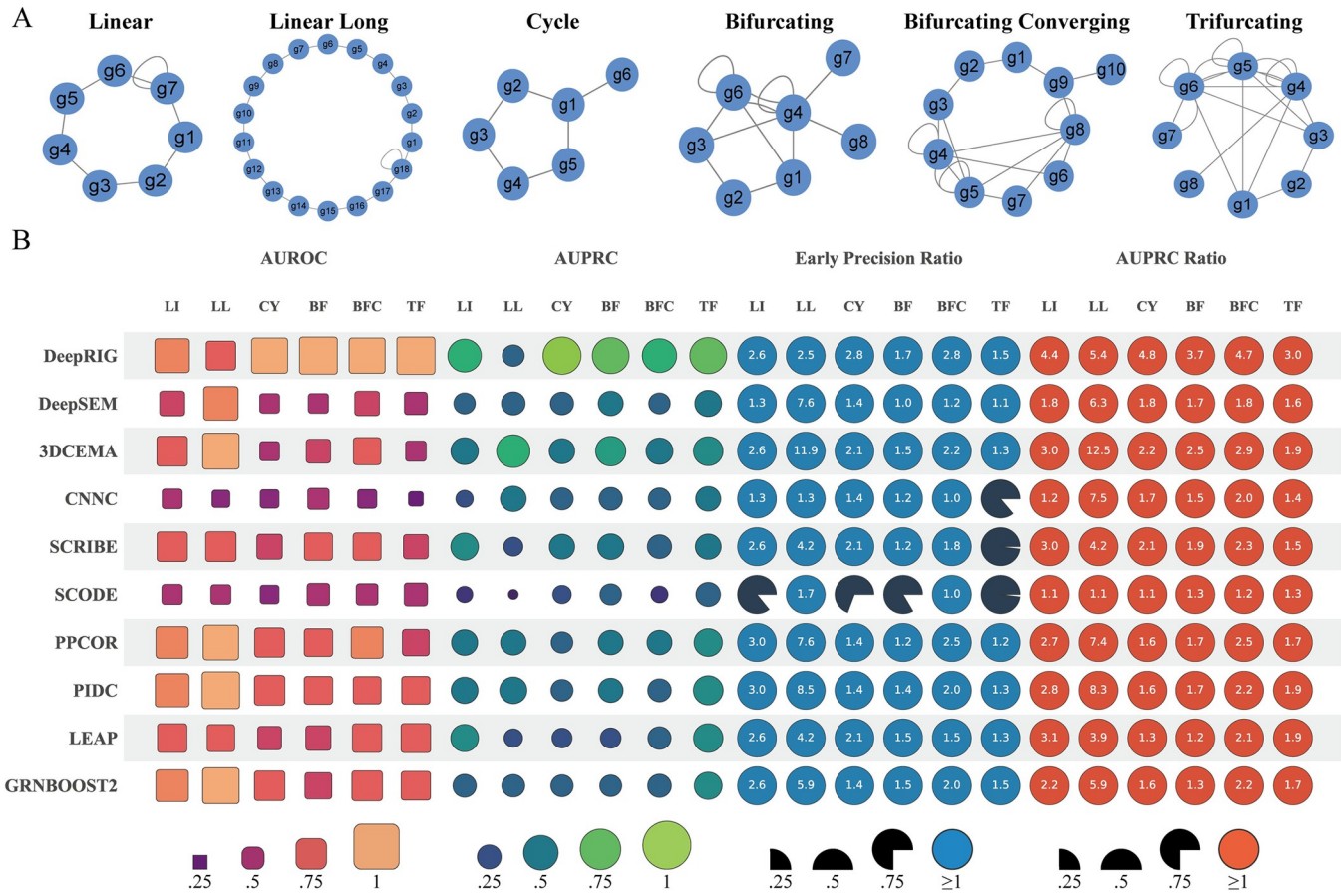

**Fig 2. Evaluation of DeepRIG on six synthetic networks.** (A) Six synthetic networks, including Linear (LI), Long Linear (LL), Cycle (CY), Bifurcating (BF), Bifurcating Converging (BFC) and Trifurcating (TF), served as ground truth in this evaluation. (B) Evaluation performance of DeepRIG and 9 state-of-the-art algorithms of GRNs in terms of AUROC, AUPRC, EPR, and AUPRC ratio. Each row represents the performance in terms of four metrics from one method in the evaluation. For the metrics of AUROC and AUPRC, the size of a shape is proportional to the corresponding AUC value, limited between 0 and 1. Regarding EPR, a circle denotes a model with an EPR value of exactly 1, indicating a result equivalent to that of a random predictor. A circle with a value denotes the model with a higher EPR value than a random predictor, while a gray wedge represents a method performing worse than a random predictor.

## Verification and comparison

**Evaluation of DeepRIG using *in silico* scRNA-seq datasets.** We first evaluated DeepRIG by using *in silico* gene expression profiles of single cells generated from six synthetic regulatory networks, which provide complete information covering varies of regulatory relationships, including Linear (LI), Linear Long (LL), Cycle (CY), Bifurcating (BF), Bifurcating Converging (BFC), and Trifurcating (TF) (Fig 2A). We considered the synthetic networks as the corresponding ground truth and compared the inferred gene regulatory networks against them. DeepRIG, CNNC, and 3DCEMA are supervised methods that train models using training data with labels that provide the expected output of the sample, allowing the model to learn and optimize by comparing the difference between its predicted output and the actual label. Therefore, we used part of the ground truth ChIP-seq data to train their models and then evaluated them in the remaining ChIP-seq data, while other unsupervised methods do not use ChIP-seq information. To some extent, this makes the comparisons not completely fair.

Fig 2B presents the inferred performance on six synthetic networks in terms of four metrics. DeepRIG had median AUROC values greater than 0.75 on all synthetic networks, and even dominated in four out of them, namely, CY, BF, BFC, and TF synthetic networks, with

achieving median AUROC values over 0.9 (S3 Table). In the LI network, the performance of DeepRIG (0.87) is comparable and slightly better than that of PIDC (0.86). In terms of AUPRC, DeepRIG had the best performances on five out of six synthetic networks, with the specific AUPRC values of LI (0.7), CY (0.93), BF (0.86), BFC (0.75), and TF (0.87), significantly outperforming other compete algorithms (S3 Table). In the LL network, the median AUPRC value of DeepRIG (0.37) was lower than that of 3DCEMA. To test the ability of inferring true positives and AUPRC values, we also explored the "top-k" inferred regulatory edges from DeepRIG, where k denotes the edge number of the ground truth. As expected, DeepRIG achieved highest EPR in CY, BF, BFC, and TF synthetic network, and second best in LI synthetic network. In terms of AUPR-R values, DeepRIG outperformed all the other GRN algorithms significantly in five out of six synthetic networks, namely, LI, CY, BF, BFC and TF. In summary, DeepRIG reached the top level of performance in LI, CY, BF, BFC, and TF synthetic network with dense regulatory relationships, indicating DeepRIG can capture the complex regulatory structures in the simulated networks.

We also sought to test the stability of inferred GRNs from DeepRIG by varying the number of cells 500, 2000, 5000 (S1 Fig). The box plots of AUROC grouping by number of cells denote that more cells contributed to better performance. Similar to the overall performance, the results of each group of cells from DeepRIG achieved top level in five synthetic networks except LL network. In addition, DeepRIG showed good stabilities in varied number of cells in these five synthetic networks.

We then evaluated the performance of DeepRIG by using *in silico* single cell datasets of four curated regulatory networks from literatures, which can capture the complex regulatory relationships in development and differentiation process of cells. The 30 *in silico* single-cell gene expression datasets generated by BEELINE for each of mCAD [23] (5 genes), VSC [24] (8 genes), HSC [25] (12 genes), and GSD [26] (19 genes) were employed in the evaluation (Fig 3A). To better mimic the zero-inflation characteristic of real single-cell gene expression profiles, three levels of dropout rates were also introduced into simulated datasets, that is, 10 datasets without dropout, 10 datasets with dropout rate 50, and 10 datasets with dropout rate 70. Then, we tested DeepRIG as well as 10 baseline algorithms using 30 different *in silico* datasets for each curated model, and evaluated the reconstructed GRNs against the corresponding curated network, which served as ground truth.

Fig 3B shows the evaluations of DeepRIG as well as 9 baseline algorithms. Similarly, we also computed the median AUROC, AUPRC, EPR, and AUPR-R values in four curated models. We observed that DeepRIG had the highest median AUROC and AUPRC values in four curated networks, with significant outperforming all other baseline methods (Fig 3B and S4 Table). One noted that in the mCAD network most baseline algorithms except SCODE had AUROC values lower than 0.5 which denotes the random predictor, while DeepRIG achieved 0.996. In addition, all baseline methods performed poorly in terms of AUPRC in the HSC and GSD curated models, while DeepRIG had AUPRC values higher than 0.8. In terms of the EPR, DeepRIG significantly outperformed all other baseline methods in VSC, HSC, and GSD networks, denoting that it could recover the true relations within the top-k inferences. All algorithms except SCODE performed poorer than the random predictor (EPR = 1), while SCODE performed slightly better than that. In terms of AUPR-R, DeepRIG showed dominating performance in four curated networks. These results denote that DeepRIG is capable to reconstruct the complex regulatory networks in four curated regulatory models.

To thoroughly assess the performance of DeepRIG in predicting activating and inhibitory regulations, we conducted an in-depth investigation on four curated networks (S2 Fig). We designed an approach where activating regulations were assigned label 1, inhibitory regulations were labeled -1, while 0 denotes no regulation. We then calculated the root mean square

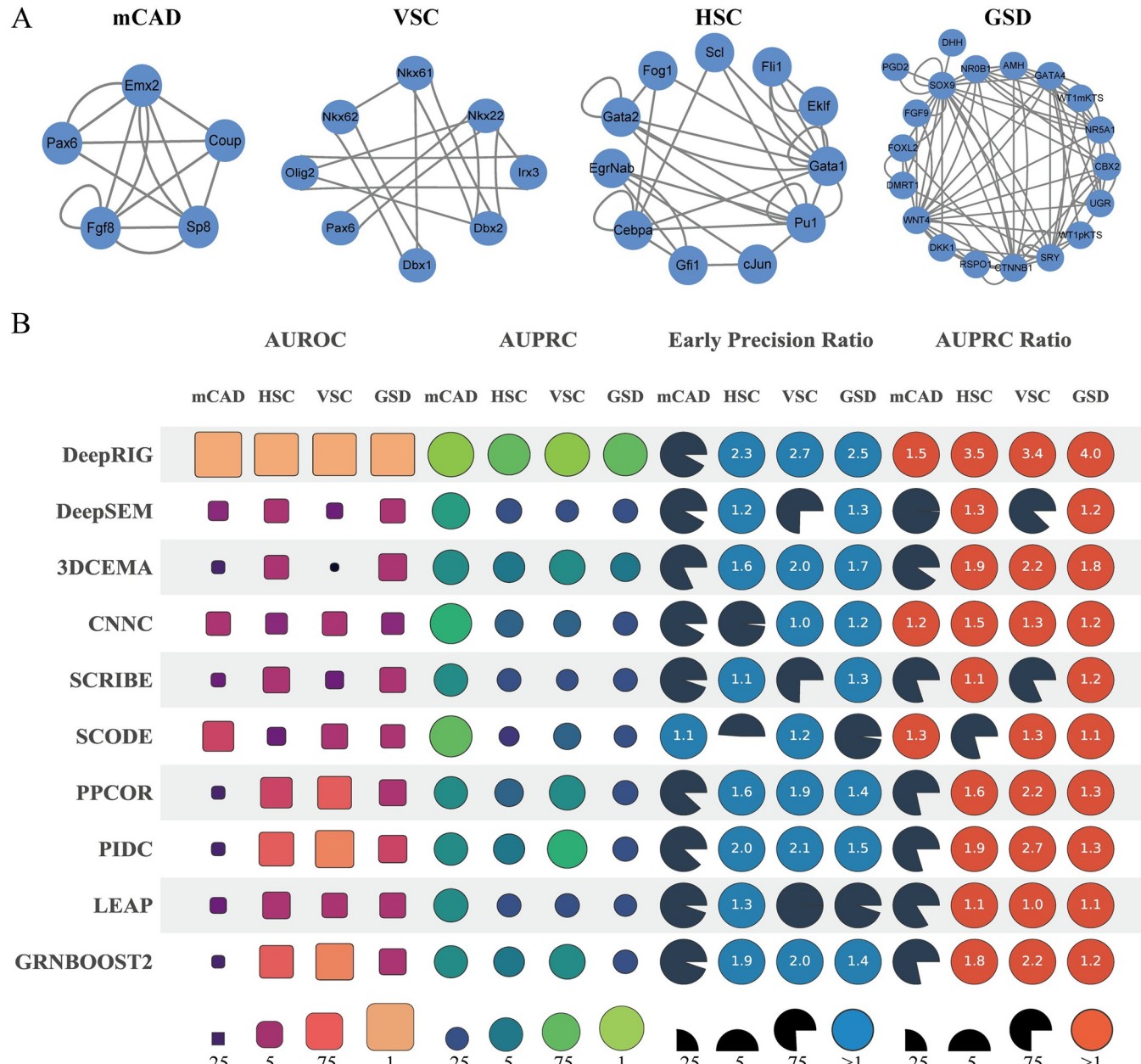

**Fig 3. Evaluation of DeepRIG on four curated networks.** (A) Four curated networks, including mCAD [23], VSC [24], HSC [25], and GSD [26], served as ground truth in this evaluation. (B) Evaluation performance of DeepRIG and 9 state-of-the-art algorithms of GRNs in terms of AUROC, AUPRC, EPR, and AUPRC ratio. Each row represents the performance in terms of four metrics from one method in the evaluation. For the metrics of AUROC and AUPRC, the size of a shape is proportional to the corresponding AUC value, limited between 0 and 1. Regarding EPR, a circle denotes a model with an EPR value of exactly 1, indicating a result equivalent to that of a random predictor. A circle with a value denotes the model with a higher EPR value than a random predictor, while a gray wedge represents a method performing worse than a random predictor.

error (RMSE) between the predicted signed GRNs and the corresponding ground truth labels. We compared DeepRIG with five baselines that also output signed regulatory networks, including PPCOR, LEAP, SCODE, SCRIBE, and CNNC. The results indicated that DeepRIG achieved the smallest RMSE on HSC, VSC, and GSD networks, and the second-smallest RMSE on the mCAD network. These findings suggest that our model outperformed the other baseline methods in predicting both activating and inhibitory regulations.

We further tested the robustness of DeepRIG for the "dropout" characteristic of scRNA-seq datasets. Fig 4 depicts the AUROC results for DeepRIG and 9 baselines within three levels of dropout rates. DeepRIG showed top level of robustness and achieved above 0.9 AUROC values on all four curated models. As the dropout rate increases, the median AUROC value of

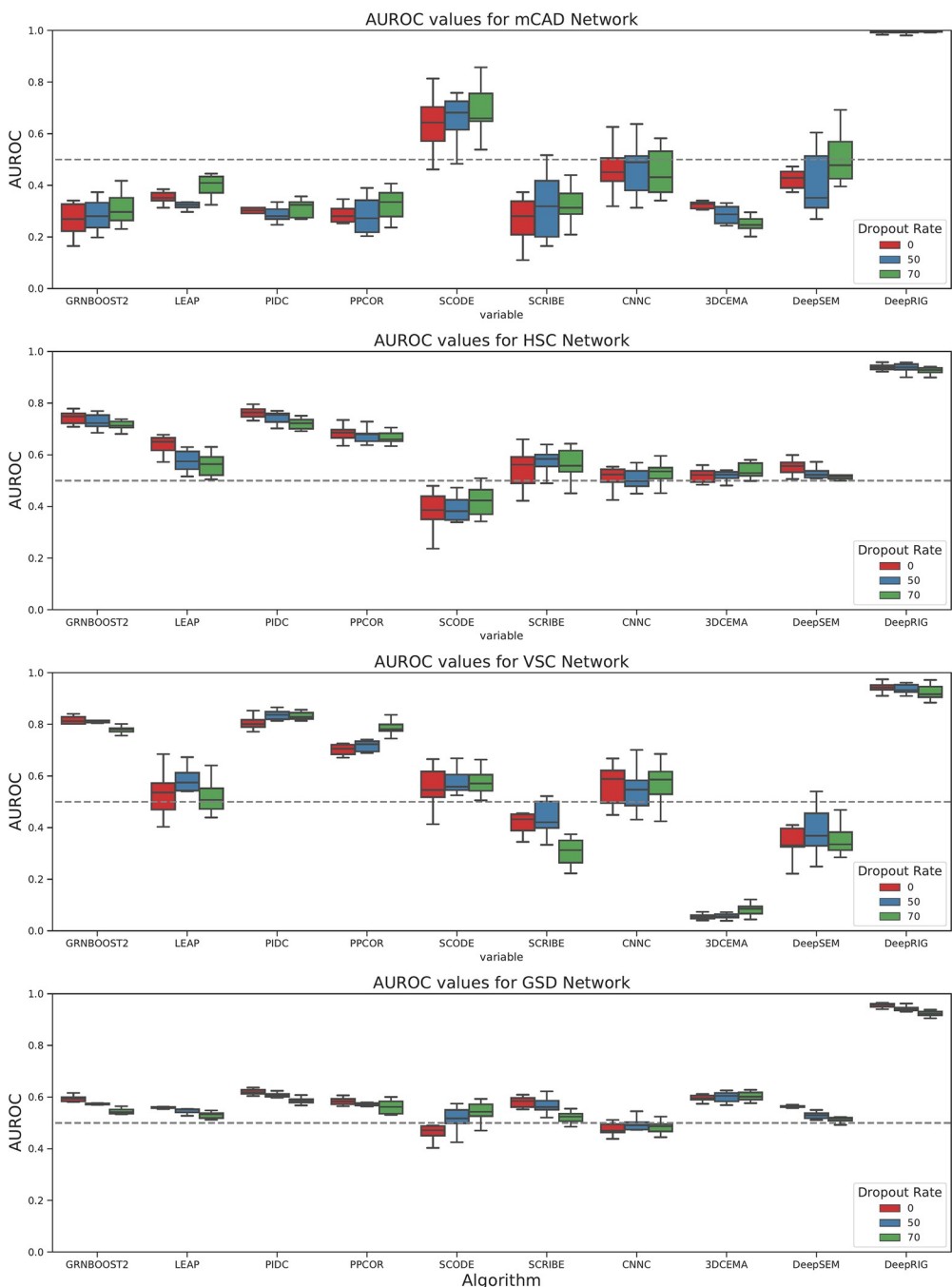

**Fig 4. AUROC boxplots on *in silico* scRNA-seq datasets from four curated networks with varies dropout rates.** Each subfigure corresponds to one of the four curated models, namely, (**A**) mCAD, (**B**) HSC, (**C**) VSC, (**D**) GSD. Each box of one method represents the values of 10 repeat runs (on 10 different datasets). Red, blue and green respectively denote to the *in silico* scRNA-seq datasets with no dropouts, a dropout rate of 50%, and a dropout rate of 70%. In addition, the gray dotted line represents the performance of the random predictor.

DeepRIG had remained almost constant on mCAD, VSC and HSC curated model. Even for the GSD curated model, its median AUROC values was only slightly reduced by 0.01 across three levels of dropout rates. For the other methods, we can see a significant change in AUROC values as the dropout rate increases. Among them, GRNBOOST2, PIDC, PPCOR, and 3DCEMA had slight fluctuations across three levels of dropout rates on four curated model. DeepRIG and PPCOR both used Spearman's rank correlations to measure the intensity of regulation between each pair of genes, and both had more stable AUROC performance on four curated model than LEAP, which applied Pearson's correlation. This comparison indicates that Spearman's rank correlation has better robustness and tolerance to technical noise of scRNA-seq data than Pearson's correlation. Overall, the above results suggest that DeepRIG has better robustness and stability for the dropout events of scRNA-seq datasets compared to the 9 baseline methods.

**DeepRIG can accurately reconstruct the gene regulation network.** Although synthetic networks provide well-defined gene regulatory rules and complete regulatory information, it would be still necessary to evaluate the ability of the models to capture complex regulatory structure in real cells. Therefore, we applied six scRNA-seq datasets [27, 28, 30, 43] from real cells to illustrate the GRN inference performance of DeepRIG. We used the ChIP-seq data [31] as the ground truth and generated a cell-type-specific referenced GRN for each real scRNA-seq dataset. We have shown the degree distributions of cell-type specific GRNs of six scRNA-seq datasets (Fig 5A). One observation is that the degree distributions of six GRNs are long-tailed. Additionally, the degrees of referenced GRNs for mESC, mHSC-L, mHSC-E and mHSC-GM are obviously normally distributed.

DeepRIG presented significant performance in TF-gene regulation inference on cell-type-specific ground truth ChIP-seq data (Fig 5B). Specifically, DeepRIG achieved improvements for AUROC of 22.5% on mESC dataset, 22.3% on mHSC-L dataset, 2.1% on mHSC-E dataset, and 20% on mHSC-GM dataset compared to the second-best method. For the mDC dataset, the AUROC of DeepRIG is comparable to that of DeepSEM and 3DCEMA, and slightly lower than CNNC. In terms of AUPRC values, DeepRIG had highest values on four benchmarks except hESC and mDC datasets, with an increase of 27.5%, 1%, 20.6%, 21.7% and 27% respectively compared to the second-best method. On the hESC dataset, the AUPRC value of DeepRIG is comparable to that of 7 competitive methods and only slightly lower than DeepSEM and SCODE. We also provide the specific AUROC and AUPRC values for DeepRIG and the 9 baselines (S5 Table).

We also investigated the accuracy of DeepRIG in identifying true regulation edges among the *top-k* predicted regulation edges. When considering the EPR values (Fig 5B) on six real scRNA-seq datasets, only DeepRIG achieved better EPR performance than random predictors (*EPR = 1*) across all datasets. DeepRIG had significant improvements by an average of 34.7%, 24.7%, 20.8%, 38.4%, 24.9%, 18.3%, 22.5%, 24.6% and 61.8% compared to DeepSEM, PIDC, PPCOR, LEAP, GRNBOOST2, SCODE, SCRIBE, 3DCEMA and CNNC respectively. Interestingly, DeepRIG outperformed 9 baselines on each of four scRNA-seq datasets where the degrees of reference GRNs conformed to a normal distribution. In particular, on the mESC dataset, DeepRIG achieved an EPR of 1.4, compared to 1.1 for the second-best method. This suggests that DeepRIG is a suitable method for scRNA-seq datasets with Gaussian distributions, which is a widely adopted hypothesis for gene expression data of single cells. Furthermore, we found a preference for DeepRIG's performance on the hESC (1.3), mESC (1.4), and mDC (1.3) datasets compared to the three datasets of mHSCs. These significant performances may benefit from the denser regulatory network in these three scRNA-seq data.

As a graph auto-encoder model, DeepRIG outperformed DeepSEM, which applies a variational auto-encoder model. The significant improvement can be attributed to the utilization of the gene prior graph constructed from the weighted gene co-expression network in our

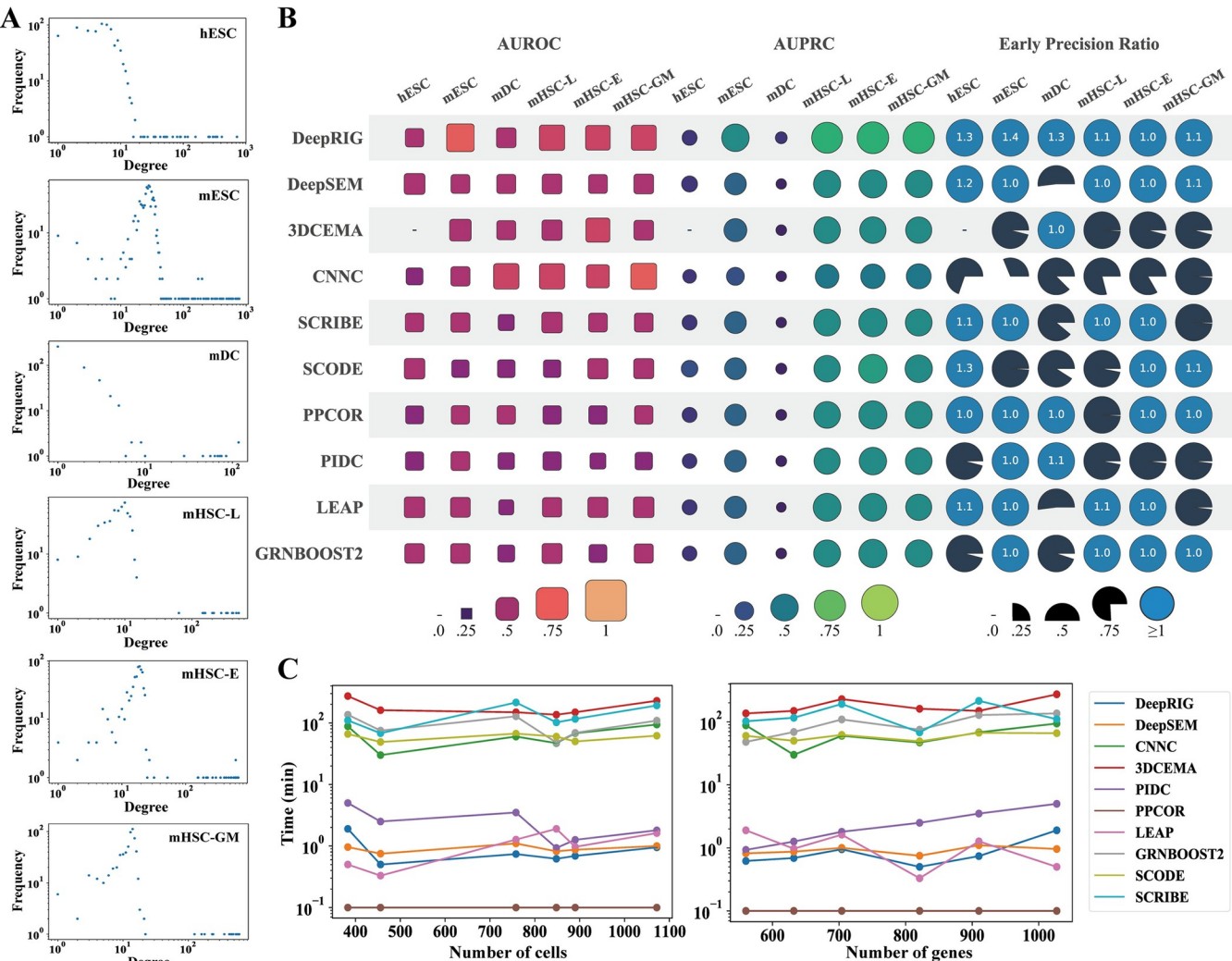

**Fig 5. Evaluation of DeepRIG on six real scRNA-seq benchmarks with ChIP-seq ground truth.** (A) Degree distributions of cell-type-specific ground truths for hESC, mESC, mDC, mHSC-L, mHSC-E and mHSC-GM. (B) Evaluation performance of DeepRIG and 9 state-of-the-art algorithms of GRNs in terms of AUROC, AUPRC and EPR. Notably, DeepRIG, CNNC, and 3DCEMA are supervised models trained on ground truth ChIP-seq data while most of the other methods are unsupervised and do not use ChIP-seq information for training. Each row represents the performance of one method across the four metrics in the evaluation. For AUROC and AUPRC, the size of a shape is proportional to the corresponding AUC value, limited between 0 and 1. Regarding EPR, a circle denotes a model with an EPR value of exactly 1, indicating a result equivalent to that of a random predictor. A circle with a value denotes the model with a higher EPR value than a random predictor, while a gray wedge represents a method performing worse than a random predictor. (C) Computational times for DeepRIG and 9 state-of-the-art GRN inference algorithms on both cell and gene numbers. All values of runtime are given in minutes.

method, which is not incorporated in the variational auto-encoder model. This comparison demonstrates crucial information provided by the gene prior graph for GRN inference. While CNNC showed comparable performance in terms of AUROC, our model significantly outperformed it in terms of AUPRC and EPR, which are more suitable metrics for evaluating imbalanced sample classification tasks like GRNs inference. Unlike AUROC, which can lead to misleading evaluations in such scenarios, AUPRC and EPR provide a more accurate assessment of the model's performance on the minority class (i.e. positive samples). Through a meticulous comparison of the three supervised learning methods, DeepRIG exhibited superior performance, proving its practical superiority over CNNC and 3DCEMA, particularly in capturing the regulatory relationships with high precision and recall.

We acknowledge that comparing supervised and unsupervised methods directly may raise concerns about fairness, as they have different access to prior knowledge. It is indeed true that supervised learning methods, such as DeepRIG, CNNC, and 3DCEMA, benefit from the utilization of ground truth ChIP-seq data, which can significantly enhance their performance in GRN inference. We humbly recognize the advantage that prior knowledge provides to supervised methods in terms of performance improvement. At the same time, it is important to highlight that this advantage does not diminish the value of unsupervised methods. Unsupervised approaches, despite lacking the benefit of ChIP-seq data, still play a crucial role in inferring GRNs and have their own merits in handling diverse biological scenarios where prior knowledge may be limited or unavailable.

Due to the varying sizes of scRNA-seq datasets, ranging from hundreds to millions, biologist also raise some concern on the computational complexity to data sizes of GRN inference methods. We then evaluated the computational complexity of DeepRIG on six scRNA-seq datasets by comparing its runtimes to those of 9 state-of-the-art algorithms. We both discussed the correlation between computational complexity and number of cells and genes. We observed three levels of the computational complexity among the 10 methods (Fig 5C). PPCOR exhibited the best computational complexity and spent only a few seconds to complete the GRN inference task. DeepRIG, PIDC, LEAP, DeepSEM are in the second level, taking a few minutes for the six benchmarks. Meanwhile, GRNBOOST2, SCODE, SCRIBE, CNNC and 3DCEMA spent hundreds of minutes to generate the GRNs. In detail, DeepRIG's computational complexity was comparable to LEAP and DeepSEM, and completed the GRN inference on six benchmarks within only one minute. We further applied our model to the whole PBMC8k dataset with 8339 cells to demonstrate its scalability in handling larger cell numbers. We also compared our model to PIDC, PPCOR, GRNBOOST2, DeepSEM, CNNC which do not require peseudotime or stamped information. As expected, PPCOR only took 1 minute to predict GRNs, and DeepRIG and DeepSEM completed the GRN inference in just 4 minutes and 10 minutes, respectively. On the other hand, PIDC and CNNC required 60 to over 100 minutes to generate the outputs, while GRNBOOST2 took thousands of minutes. This comparison clearly demonstrated the scalability of our model when dealing with datasets containing a large number of cells. Regarding gene scalability, we evaluated our model and five baselines on the TNBC scRNA-seq dataset containing 7185 genes. Notably, PPCOR completed predictions in 22 minutes, while DeepRIG and DeepSEM took 61 and 140 minutes, respectively, for GRN inference. GRNBOOST2 required 1383 minutes, while CNNC and PIDC failed to provide predictions. Additionally, we observed that the time complexity of most methods is primarily influenced by the number of genes. This comparison suggest that DeepRIG is a computation-efficient algorithm and highlights its scalability in handling datasets with a large number of cells and genes.

To ensure what portion of all the interactions should be used to train the model for making more accurate predictions, we further investigated on mESC scRNA-seq dataset. We used different portions of all the ground truth interactions to train our model and utilized the rest to test the model (S3 Fig). We found that DeepRIG reached highest EPR median value when using 67% ground truth interactions for training, compared to other portions.

**Incorporating graph autoencoder improves inference performance of GRNs.** As a popular and powerful algorithm for node classification and link prediction, GCNs are capable of learning the spatial-location structures and node embeddings of a graph. To investigate the role played by GAE and whether it could improve the performance in the task of GRN inference, we compared DeepRIG with the straightforward WGCNA model that simply calculated the Spearman's correlation coefficients and did not apply GAE model. The gene co-expression network from the WGCNA model was denoted as gene regulation network and evaluated by

the same standards as DeepRIG. LEAP computes the maximum Pearson's correlation coefficients as the regulatory score for each pair of genes. In addition, PPCOR is also a correlation-based method that applying Spearman's rank correlations as the confidence of potential interaction. So, we also consider these two methods into the comparison.

The evaluation results for four models in comparison are presented in Fig 6. Comparing the WGCNA model, DeepRIG improves the performance of GRN inference at 15.3%, 45%, 15.3%, 10.9%, 10%, and 11.2% on six benchmarks in terms of EPR values. The results indicate that DeepRIG can correctly predict more regulation relationships that verified by the ground truth. For AUROC and AUPRC, DeepRIG reaches the average improvements at 42% and 33%, respectively. Notably, DeepRIG poses significant improvements on those datasets (hESC, mESC and mDC) which contain more genes and cells in terms of EPR. The result shows the differences in inference performance between the WGCN and the final GRNs inferred by DeepRIG. These comparisons clearly demonstrate that GAE model has the capability to learn the global regulatory structure from the prior regulatory graph (a.k.a. WGCN) and identify the latent regulation patterns, thereby enhancing the performance of GRN inference. Another interesting result observed from the figure is that the WGCNA model performed comparable to LEAP and PPCOR in terms of all three metrics. It suggests that the WGCN containing global regulatory information based on correlation coefficients can reflect gene regulation modes.

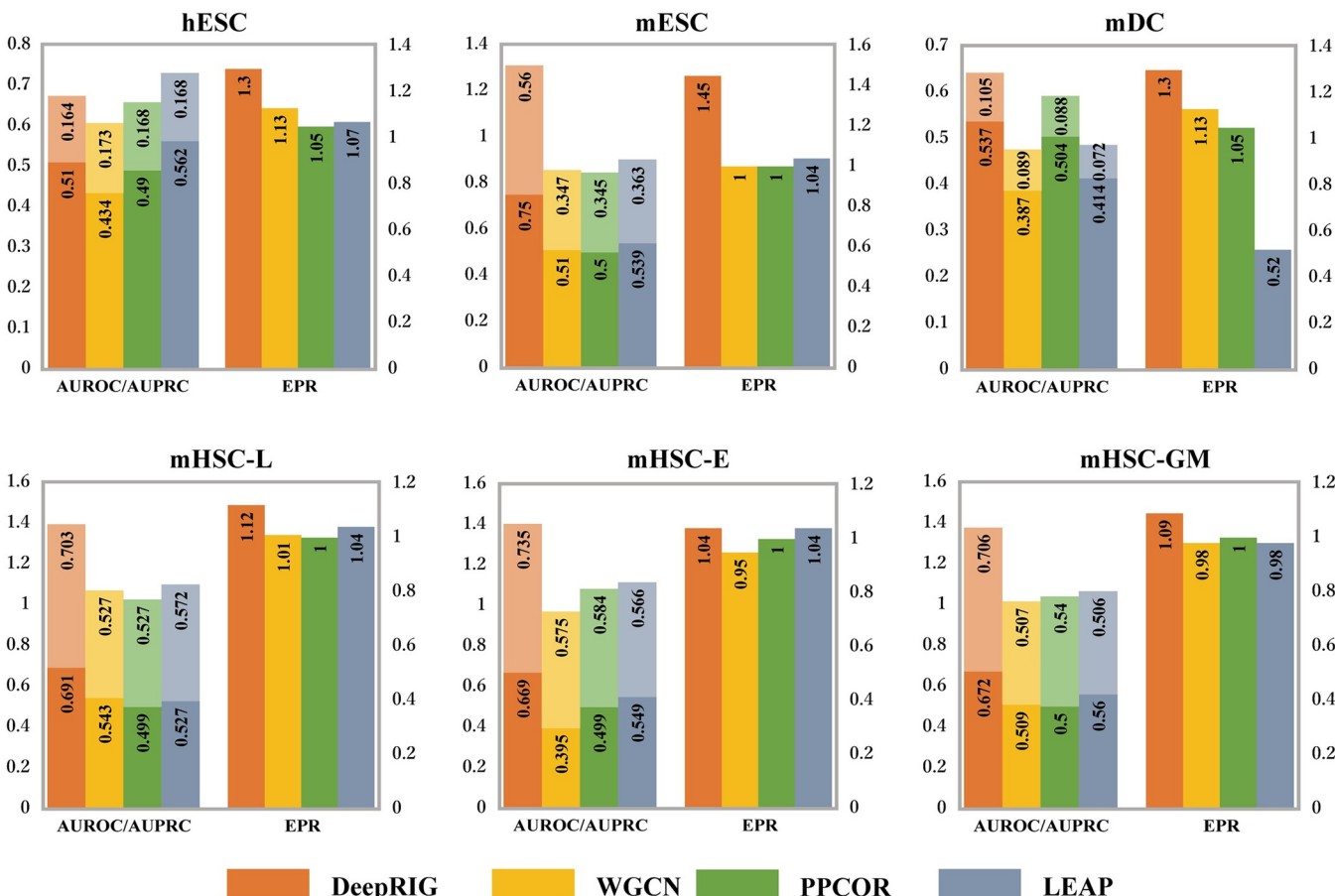

**Fig 6. Graph autoencoder model improves the performance for GRN inference.** The figure shows the results for DeepRIG, straightforward WGCNA model, PPCOR and LEAP on six benchmarks. The dark shade of the same color represents AUROC while the light shade represents AUPRC.

## Applications

**Using DeepRIG for inferring cell-type specific GRNs from PBMCs.**   Cell-type specific GRNs define the cell transcriptional states during the development. The cell type can be defined by a particular set of active transcription factors (TFs). Cell-type specific GRNs provide an unbiased perspective for examinations. To further validate the abilities of cell-type specific GRN inferring of DeepRIG, we applied it on the scRNA-seq samples of more than 8000 human peripheral blood mononuclear cells (PBMCs8k) derived from 10X Genomics. After data preprocessing and clustering with scanpy [32] pipeline, we manually annotated cell types by searching the literature to identify marker genes. As a result, we focused on four cell clusters with the highest cell numbers, namely CD4+ T cells, CD14+ Monocyte cells, CD8+ T cells, and B cells, for cell-type-specific GRN inference (Fig 7A). For the ground truths of PBMCs, we downloaded the gold standard networks from hTFtarget [44] database which integrated ChIP-seq, TF-binding sites and epigenetic modification information. For each cell type, we trained the DeepRIG model and applied it to infer the cell-type specific GRNs. For the inferred GRNs from DeepRIG, we kept those regulations with edge weights greater than 0.2. We identified and counted the regulatory interactions that were newly predicted by DeepRIG, as well as those that were already included in the hTFtarget interactions used during training (S6 Table). We observed that no more than 30% of the predicted regulatory relationships were prior knowledge used for training, while the remaining 70% of the regulatory relationships were newly predicted by DeepRIG. Furthermore, we ordered the genes by their degrees and identified those genes whose outdegree were higher than 20 as core TFs.

As a result, we obtained the reconstructed GRNs of four cell type with 33, 1914, 511, 190 regulations, respectively. In CD14 + monocyte cells, we first screened 19 core TFs among which CEBPB [45] and SPI1[46, 47] have been identified critical TFs for monocytic cells of proliferation and differentiation. We observed complex regulatory subnetworks of CEBPB and SPI1 in CD14+ monocytes, but only one regulation or very simple regulation structures in other three cell types (Fig 7B). In the regulatory subnetwork of CEBPB, we further examined the regulated genes and observed four genes including CD68, TYROBP, LST1, S100A11, that have been identified as marker genes of CD14+ monocytes (Fig 7C). We then examined the expression of four genes, they were highly expressed in CD14+ monocytes but lowly expressed in other cell types (Fig 7D). Similarly, we explored the regulatory subnetwork of SPI1, and observed two marker genes of CD14+ monocytes, namely LGALS2 and PSAP (Fig 7C). We have also identified GATA2 and STAT1 as two specific regulators that exhibit cell-type-specific regulatory structures in CD14 Monocytes. GATA2 demonstrated 201 degrees in CD14 Monocytes, while only 3 degrees in B cells. Furthermore, GATA2 exhibited the highest Betweenness Centrality value (0.59), indicating its significant influence and control over other genes. GATA2 is recognized as an important TF involved in regulating various biological processes, including ontogeny in monocytes [48]. Similarly, STAT1, which showed sharp degree changes across cell types, has been reported as a critical factor driving peak pSTAT1 levels in CD14 + monocytes [49]. In B cells, one of the central TFs, namely MEF2C, was also identified as a marker gene of B cells (Fig 7C). As a calcineurin-regulated TF in B cells, MEF2C is required for B-cell proliferation, and its expression is specific to receptor stimulation [50]. Another central TF, EGR1, has been identified as a critical TF for B-cell development and immune response [51]. In the regulatory subnetwork of EGR1, two target genes, IL4R and BIRC3, were highly expressed in B cells and lowly expressed in other cell types (Fig 7D), while IL4R was also identified as a marker gene of B cells (Fig 7C). PAX5 and RUNX3 were also identified as a cell-type-specific regulator by our model. Indeed, the well-established transcription factor PAX5 plays a pivotal role in governing the identity and function of B cells throughout B

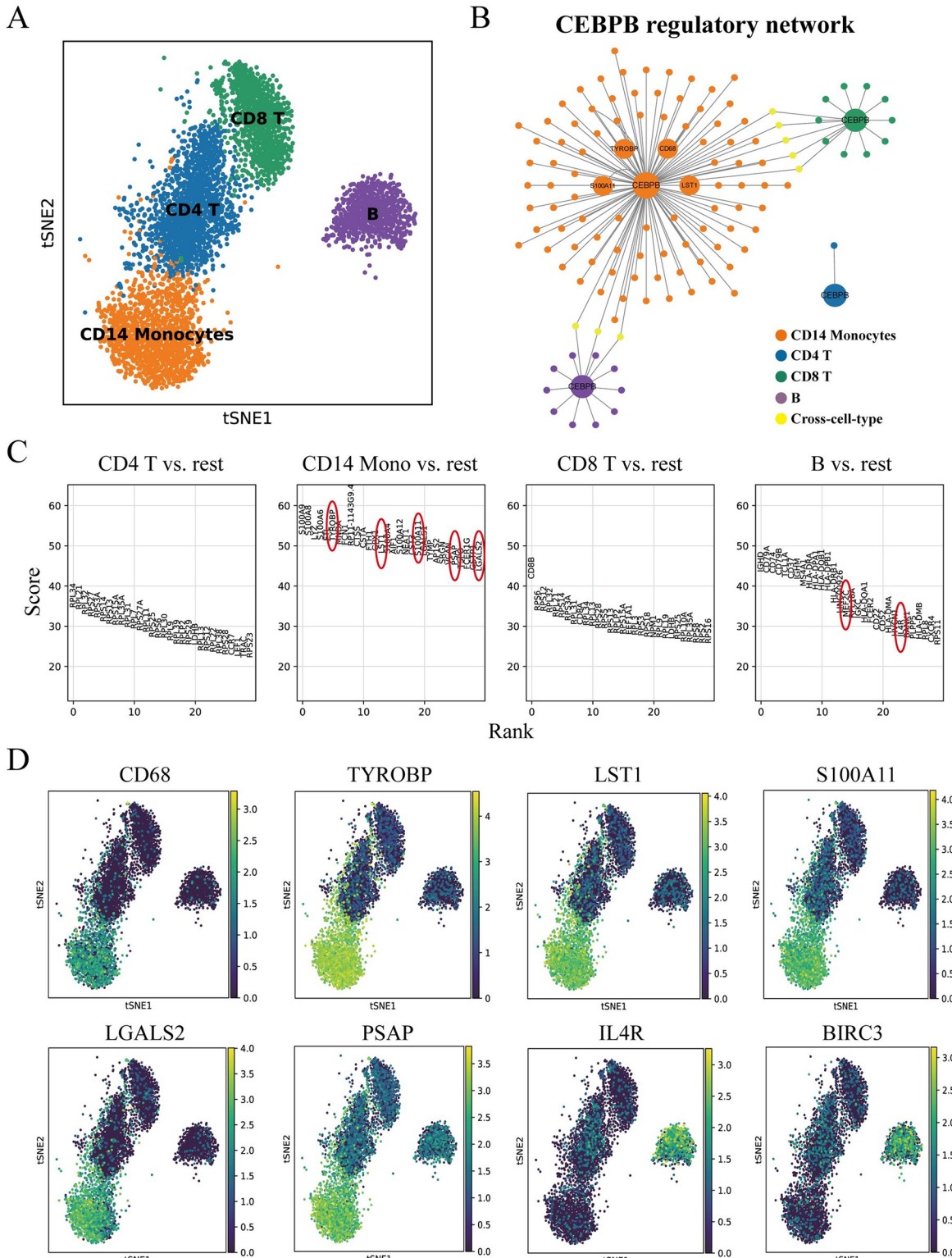

**Fig 7. Inferring cell-type specific GRNs by DeepRIG enables to assist in identifying the cell type of PBMCs.** (A) The tSNE visualization of 4 cell types (CD4 T, CD14+ monocytes, CD8 T and B cells) of PBMCs with highest cell numbers. (B) The identified core TF, CEBPB, have a complex regulatory network in CD14+ monocytes but only on regulation or simple networks in other cell types. (C) 30 marker genes were identified for 4 cell types. (D) tSNE visualization of four cell types in the analysis colored by expression levels of several marker genes including CD68, TYROBP, LST1, S100A11, LGALS2, PSAP, IL4R, and BIRC3.

lymphopoiesis and has been associated with human B cell malignancies [52]. RUNX3 is known for its cross-regulation mechanism with RUNX1 in B cells, and its expression is required for efficient B cell proliferation [53]. The comprehensive list of identified regulators specific to each cell type can be found in S7 Table. The results suggest that DeepRIG provides cell-type-specific GRNs predictions and the inferred GRNs can further contribute to identify marker genes for cell type annotations.

We conducted further investigations to assess the transferability of our model across different cell types. We trained our model on CD14 Monocytes and subsequently transferred it to three other cell types. And we found that our model presents good transferability across different cell types when comparing the predicted results from transferring to those obtained from each cell type from scratch. For instance, in the case of B cells, we calculated the Root Mean Square Error (RMSE) between the predicted results obtained from transferring and those derived from training B cells from scratch (S5 Fig). Notably, the RMSE between the two models within B cells was significantly lower compared to the RMSE between the transferred results from B cells and the results of CD14 monocytes trained from scratch, as well as the results of the other two cell types trained from scratch. This analysis suggests that the transferred results from B cells align more closely with the predicted results obtained from training B cells from scratch, rather than being heavily influenced by the training on CD14 monocytes. Furthermore, we have provided a comprehensive list of the identified regulators from both the transferred model and the model trained from scratch (S8 Table).

We then explored the differences between the final inferred GRNs and the WGCN from CD14 Monocytes and CD8T cells in PBMCs (S6 Fig). As expected, they showed contrasting network topologies in visualizations and the distributions of various topological attributes. Our observations revealed distinct characteristics of the final inferred GRNs, including the presence of sub-regulatory networks with central regulators, which were not observed in the WGCN. Furthermore, significant differences in the distributions of several topological attributes such as degree, closeness, hub score, and edge betweenness were observed between the WGCN and the final inferred GRNs. The visualization and analysis of these network structures and topological attributes suggest that the final inferred GRNs are more effective in identifying hub genes or key regulators.

We compared the performance of several other GRN inference methods, including PIDC, PPCOR, GRNBOOST2, DeepSEM, and CNNC, with our proposed DeepRIG method on CD14 Monocytes of PBMCs. Visualizing and analyzing the network structures of the inferred GRNs from these methods, as well as DeepRIG, revealed interesting findings (S7 Fig). We observed that the GRNs inferred by CNNC and GRNBOOST2 exhibited similarities to the predictions of our model, albeit with a more dispersed network structure. On the other hand, the GRNs predicted by the correlation-based methods, PIDC and PPCOR, were found to be more akin to the WGCN. Notably, significant differences were observed among these methods in terms of topological attributes, such as degrees, closeness, hub scores, and edge betweenness. A particularly intriguing observation was that the inferred GRNs from our model and CNNC exhibited a power-law distribution, resembling scale-free networks. In scale-free networks, a few key hub nodes have rich connections to other nodes, while most nodes have limited connections. This distribution pattern is commonly observed in various real biological networks, including metabolic networks [54, 55], protein-protein interaction networks [56], and transcriptional regulation networks [57]. Our analysis suggests that the GRNs inferred by our model better capture the regulatory relationships present in the real data and are more effective in identifying hub genes or regulators.

**DeepRIG can infer novel regulators of TNBC cells.** To validate whether DeepRIG can infer novel regulators, we further applied it to predict the GRN from the scRNA-seq data of

triple-negative breast cancer cells (TNBC). As an aggressive subtype of breast cancer that accounts for approximately one-fifth of all breast cancers, TNBC is characterized by extensive intratumoral heterogeneity and exhibits a more aggressive clinical course compared to other subtypes. In this study, we downloaded the scRNA-seq data of 1534 cells from six untreated primary TNBC tumors as well as the normal breasts [58]. After quality control and removing the low-expressed genes, we derived the normalized expression data with a total of 13280 genes. We also used the gold standard network from hTFtarget database [44] as the ground truth of the breast model. We removed those TF regulation associations in tissues other than mammary glands. A total of 14915 regulation entries remained after filtering. We used the scRNA-seq data of normal breast and gold standard network to train DeepRIG. After training, we used the trained model to infer the GRN of scRNA-seq data from TNBC tumors. For the inferred GRN from DeepRIG, we retained those regulatory associations with weights greater than 0.2. In a result, we reconstructed the GRN of TNBC with 173,399 regulatory associations among 7185 genes (Fig 8A). Furthermore, we counted the number of degrees of each node in the inferred GRN, and calculated their betweenness centrality and PageRank scores (Fig 8B), two important network analytics metrics that assess the importance or influence of nodes in a network. Specifically, betweenness centrality measures the intermediacy of a node in the network, while PageRank measures the global importance of a node based on the network's link structure and importance propagation. Genes with high betweenness and PageRank scores are more likely to be considered as important hub genes or regulators.

From the reconstructed GRN and its topology analysis, we observed ten statistically significant genes comprising ARF1, CD9, DSTN, ENO1, YWHAQ, RPS5, EEF1G, RPL5, ATP5B, and KRT8 (Fig 8). Interestingly, these genes with top degrees (colored) follow a circular shape at the center of the regulatory network (Fig 8A), indicating that they play a crucial role in GRN of TNBC patients, such as hubs and bottlenecks. Further, four of these ten genes have been confirmed by several literatures to play a vital role in the development of TNBC and related therapeutic drugs. Particularly, both CD9 and ARF1 show the most degrees, the highest betweenness centrality and PageRank (Fig 8B), which indicate they are the main hubs and bottlenecks in the GRN of TNBC. Practically, CD9 has been reported to a potential biomarker for prognosis in breast cancer [59, 60, 61]. Hee et al. [61] found that the expression of CD9 in tumor cells leads to a worse prognostic survival rate, while the expression in stromal immune cells significantly correlates with a better prognostic survival rate. ARF1, a part of the AMPK pathway that regulates the autophagic cell death in breast tumor, is considered a potential crucial regulator in TNBC and a promising therapeutic target for the treatment [62]. ENO1, another gene that was founded to have significant characteristics of topology, has been reported as a functional target of a microprotein TRPC5 opposite strand that plays an important role in TNBC progression, suggesting its potential application as a new therapeutic target and a biomarker for prognosis in TNBC [63, 64]. Overexpression of KRT8 has also been reported in autophagy-deficient mammary epithelial cells as well as breast tumors, making it be an appropriate marker for the functional status of autophagy in human breast cancers, and may have promising applications in the treatment of patients with breast tumors [65, 66]. Moreover, we performed gene ontology (GO) enrichment analysis by applying ClueGO [67] to identify the function of the 179 genes with degree more than 1000 (top 11.8%). The enriched GO terms confirm their importance in TNBC pathophysiology (Fig 8C), such as the term of "interleukin-12-mediated signaling pathway" ($\rho < 8.6e-6$), which has been reported to be involved in the prevention mechanisms of breast tumor [68]. These finding demonstrate that DeepRIG can be used to obtain potential regulators or targets of the development and treatment of diseases from the inferred GRNs.

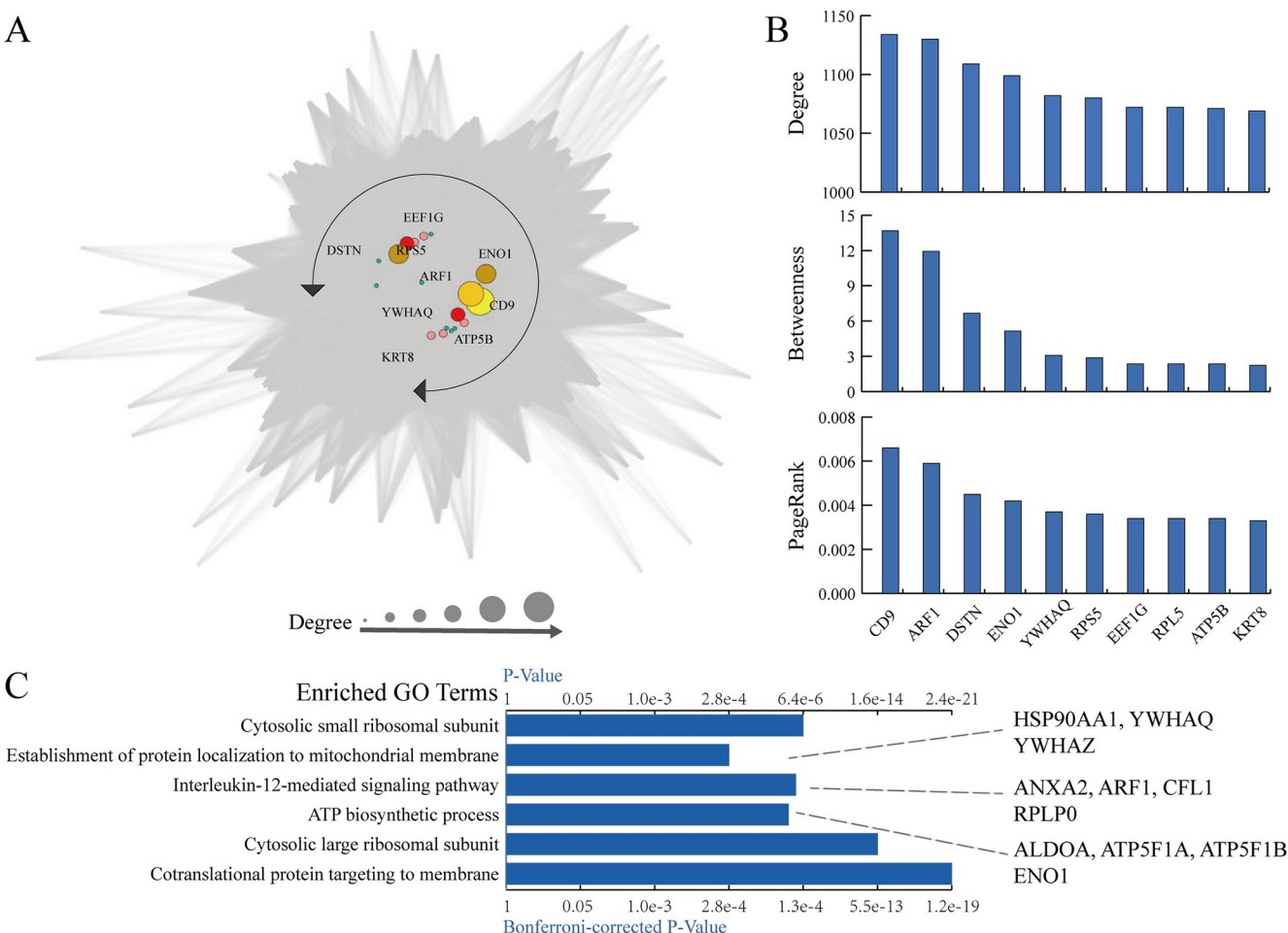

**Fig 8. Reconstruction of regulatory network in triple-negative breast cancer can identify new potential regulator.** (**A**) Visualization of the inferred gene regulatory network. The size of the nodes in the network is proportional to the number of degrees. The genes with top degrees (colored) follow a circular shape at the center of the regulatory network, and become important bottlenecks. (**B**) The top 10 genes in terms of degrees, betweenness centralities, and PageRank scores are represented in three histograms. Betweenness centrality and PageRank score are two important network analytics metrics that assess the importance or influence of nodes in a network, and genes with high betweenness and PageRank scores are more likely to be considered as important hub genes or regulators. CD9 and ARF1 both display significant characteristics of topology, indicating their topological importance in the regulatory network. (**C**) Gene ontology enrichment analysis of genes with degrees greater than 1000 (Top 11.8%) in the GRN shows overrepresentation of TNBC-related functions.

Breast cancer, a complex and heterogeneous disease, involves significant genetic and regulatory alterations within cancer cells compared to normal breast cells. We then investigated the overlap between the inferred GRNs and the interactions used for training the model (S8 Fig). About 37.5% of the ground truth interactions (5598 interactions) used for model training were present in the final inferred GRNs (S8A Fig). Furthermore, we found that most of the newly inferred regulations are assigned the same or higher predicted scores compared to the overlapped interactions, indicating greater confidence in the newly predicted associations (S8B Fig). Among the newly predicted interactions, an interesting finding was the regulation of the HIF1α pathway by XBP1. This prediction was validated by Chen et al.[69] through a series of experiments, including invasion assays, subcutaneous xenograft experiments, motif analysis on in-house ChIP-seq data, survival analysis, RT PCR analysis and immunoblot analysis. Furthermore, DeepRIG inferred that ZNF217 regulates the expression of KLF4, FTO, and ALKBH5, and is, in turn, regulated by HIF1α. Remarkably, Zhang et al.[70] performed RT-

qPCR analysis, immunoblot assay and immunohistochemical analysis, and demonstrated that ZNF217 co-interacts with ALKBH5, functioning as a mediator to control the expression of KLF4 and NANOG involved in the post-transcriptional mechanism of HIF1α. The findings emphasize the critical role of these altered regulatory interactions in the development and progression of breast cancer.

We further compared the distributions of network topologies between the final predicted GRNs and WGCN of the TNBC scRNA-seq dataset. The results showed significant differences in node degree, closeness, hub score, and edge betweenness (S9 Fig). As expected, the distribution of network topologies from the final inferred GRNs exhibited long-tailed characteristics. Ten potential regulators identified in the final inferred GRNs (Fig 8) showed sharp degree changes compared to those in WGCN. In addition, genes including ZNF217, CREBBP, XBP1, RARG, RXRA, E2F4, RAC3, PBX1, and ELF1, also exhibited significant degree changes.

## Discussion

We proposed DeepRIG, a graph-based deep-learning model for GRNs inference from scRNA-seq profiles. To learn the global regulatory structure among genes, DeepRIG takes the trick of measuring the Spearman's correlation coefficients from the gene expression of scRNA-seq data to construct the weighted gene co-expression network, which can be further transformed into a prior regulatory graph. Then, a GAE model is used to capture the global regulatory dynamics and reconstruct the GRNs from the gene prior regulatory graph. Extensive analysis of GRNs inference on scRNA-seq datasets with golden standards both from synthetic networks and real cells demonstrates that DeepRIG reliably yields GRN inference and can accurately reconstruct the gene regulatory structures from scRNA-seq data. DeepRIG also provides the cell-type specific GRNs inference. By applying it on the scRNA-seq data from PBMCs, we identified several critical TFs and marker genes of CD14+ monocytes and B cells, which enable to help determine the cell types in turn. We also identified ten statistically significant genes by applying DeepRIG on scRNA-seq data from TNBC cells, and four of them have been confirmed by literatures to play crucial roles in the development of TNBC and related therapeutic drugs. All of these analyses suggest that DeepRIG can accurately infer the cell-type specific GRNs and identify novel regulators of progression and inhibition.

By transforming the gene expression profiles of scRNA-seq data into a weighted gene co-expression network, DeepRIG has increased robustness and tolerance to technical noise, such as false zeros from dropout events. The WGCN, also called a prior regulatory graph, contains the global regulatory structure that is crucial to identify the regulatory relationships in the complex biological systems. As a supervised method, DeepRIG can take advantage of the difference between positive and negative samples to fine-tune the parameters in GCNs and the scoring function. Extensive analysis of several GRNs inference tasks on different datasets indicates DeepRIG improves the accuracy of GRNs inference and outperforms 7 unsupervised methods and another 2 supervised in comparison. Though DeepRIG needs a fraction of ground truth to train the model, we have demonstrated that DeepRIG has good performances and outperforms other state-of-the-art methods in terms of AUROC with ChIP-seq data serving as ground truth, which is easily accessed and can be applied to most datasets. In this scenario, DeepRIG is a better method with high accuracy than other unsupervised approaches. Finally, DeepRIG can be easily extended to multiple fields of single-cell analysis. For example, DeepRIG can be further extended to model the cell-cell communications by joint exploring the protein-to-DNA and protein-to-protein regulatory interactions of different cell types.

However, our method presents several limitations. Although DeepRIG is easily applied and adaptable as long as ChIP-seq data is available, it is unable to guide the model training in the

absence of specific ground truth data. Additionally, using only scRNA-seq data as a basis for inferring GRNs can provide information on intracellular gene expression levels, it may not be sufficient for understanding the specific mechanisms of gene regulation. Therefore, the introduction of multi-omics single-cell data, especially scATAC-seq data, can provide more comprehensive and detailed information to help more accurately infer the topology and mechanisms of GRNs [71, 72]. The two limitations mentioned above will be the focus of our future work.

## Supporting information

**S1 Fig. AUROC performance on in silico scRNA-seq datasets from six synthetic networks with limited cells.** Each subfigure corresponds to one of the six synthetic networks. Each box of one method represents the values of 10 repeat runs. Red, blue and green respectively denote to the *in silico* datasets with 500, 2000, and 5000 cells. The gray dotted line represents the performance of the random predictor.
(TIF)

**S2 Fig. Performance of inferring activating and inhibitory regulations on four curated networks.** The value of the Y-axis represents the Root Mean Square Error (RMSE) between the predicted signed GRNs and the ground truth. The bar charts for each color represent the performance of a method.
(TIF)

**S3 Fig. EPR performance with different portion of all the ground truth used for training.** The portion of all the ground truth for training is 0.1, 0.2, 0.33, 0.5, 0.67, 0.8 and 0.9 in the seven columns as indicated by the X-axis labels. We performed five repeated experiments on the mESC dataset for each portion, with different random seeds used in each repetition.
(TIF)

**S4 Fig. EPR performance with different threshold values for building weighted gene co-expression networks.** The threshold values for building WGCN are 0, 0.2, 0.4, 0.6, and 0.8 in the five columns as indicated by the X-axis labels. We performed five repeated experiments on the mESC dataset for each threshold value, with different random seeds used in each repetition.
(TIF)

**S5 Fig. Comparison between Transferred and From-Scratch Models for Four Cell Types.** The value of the Y-axis represents the Root Mean Square Error (RMSE) between the predicted results from transferred models and the inferred results from trained models from scratch. The bar charts for each color represent a cell type.
(TIF)

**S6 Fig. Contrasting network topologies presented by the final inferred GRNs and the WGCN in PBMCs.** Two cell types, CD14monocytes and CD8T cells, are investigated respectively. In each subgraph, top and bottom represent the final predicted GRNs and WGCN, respectively. Left and right represent the visualization and the distribution of several topological properties of the network, respectively.
(TIF)

**S7 Fig. Difference of inferred GRNs by several different GRNs inference methods on CD14 Monocytes.** The visualization (left) and the distribution of several topological properties (right) of the inferred GRNs by (A) DeepRIG, (B) CNNC, (C) GRNBOOST2, (D) DeepSEM,

(E) PPCOR and (F) PIDC, respectively, revealed differences.
(TIF)

**S8 Fig. Statistics of the regulations in the final inferred GRNs.** (A) The overlap between the inferred regulations and the ground truth regulations used for training. (B) The distributions of the regulatory scores from newly inferred regulations and the overlapped regulations.
(TIF)

**S9 Fig. Contrasting network topologies presented by the final inferred GRNs and the WGCN in TNBCs.** Left and right columns represent the distribution of four topological properties of the final inferred GRNs and the network from WGCN, respectively.
(TIF)

**S1 Table. Summary of six scRNA-seq datasets from real cells.**
(XLSX)

**S2 Table. Summary of default hyperparameters for DeepRIG.**
(XLSX)

**S3 Table. AUROC and AURPC values on *in silico* scRNA-seq datesets from six synthetic networks.** The bold value denotes the best AUROC/AUPRC among 10 competing GRN inference methods on each benchmark, and underlined is the second-best.
(XLSX)

**S4 Table. AUROC and AURPC values on *in silico* scRNA-seq datesets from four curated networks.** The bold value denotes the best AUROC/AUPRC among 10 competing GRN inference methods on each benchmark, and underlined is the second-best.
(XLSX)

**S5 Table. AUROC and AURPC values for DeepRIG on six benchmarks based on the ground truth from ChIP-seq data source.** The bold value denotes the best AUROC/AUPRC among 10 competing GRN inference methods on each benchmark, and underlined is the second-best.
(XLSX)

**S6 Table. Predicted regulatory interactions by DeepRIG in four cell types of PBMCs: Newly predicted vs. Pre-existing in training.** The GRNs inferred by DeepRIG were subjected to edge weight normalization, ranging from 0 to 1. Subsequently, we retained the regulations with edge weights exceeding 0.2.
(XLSX)

**S7 Table. Identified regulators specific to four cell types of PBMCs by our model.**
(XLSX)

**S8 Table. Comparison of identified regulators from three cell types between trained models from scratch and transferred models.**
(XLSX)

## Author Contributions

**Conceptualization:** Jiacheng Wang, Quan Zou.

**Data curation:** Jiacheng Wang, Yaojia Chen.

**Formal analysis:** Jiacheng Wang.

**Funding acquisition:** Quan Zou.

**Investigation:** Jiacheng Wang.

**Methodology:** Jiacheng Wang.

**Software:** Jiacheng Wang.

**Supervision:** Quan Zou.

**Validation:** Jiacheng Wang, Yaojia Chen.

**Visualization:** Jiacheng Wang.

**Writing – original draft:** Jiacheng Wang.

**Writing – review & editing:** Jiacheng Wang, Yaojia Chen, Quan Zou.

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
