## [Decision Letter · Decision Letter 0]

21 Jun 2023

Dear Dr Quan Zou,

Thank you very much for submitting your Methods entitled 'Inferring gene regulatory network from single-cell transcriptomes with graph autoencoder model' to PLOS Genetics.

The manuscript was fully evaluated at the editorial level and by independent peer reviewers. The reviewers appreciated the attention to an important problem, but raised some substantial concerns about the current manuscript. Based on the reviews, we will not be able to accept this version of the manuscript, but we would be willing to review a much-revised version. We cannot, of course, promise publication at that time.

If you decide to revise the manuscript for further consideration at PLOS Genetics, please aim to resubmit within the next 60 days, unless it will take extra time to address the concerns of the reviewers, in which case we would appreciate an expected resubmission date by email to plosgenetics@plos.org.

We are sorry that we cannot be more positive about your manuscript at this stage. Please do not hesitate to contact us if you have any concerns or questions.

Yours sincerely,

Xiaojie Qiu

Guest Editor

PLOS Genetics

David Balding

Section Editor

PLOS Genetics

Editor Comments:

We thank you for your submission. After careful consideration and evaluation by three reviewers and our editorial team, we believe that your manuscript, "Inferring gene regulatory network from single-cell transcriptomes with graph autoencoder model", will benefit from a major revision to address several important concerns raised.

In particular, Reviewer 1 acknowledges the novelty of your method, DeepRIG, for GRN inference but highlights several issues that need clarification. The most significant concerns are related to the comparison between the WGCNA network and the final GRN, as well as the perceived unfair comparison to other methods, especially in Figure 5B. The reviewer suggests that a supervised model trained on ground truth ChIP-seq data could inherently perform better than unsupervised methods, and this needs to be clearly stated in your manuscript. Furthermore, the incorporation of directionality and inhibitory relationships could potentially improve the manuscript's value. Reviewer 2, while recognizing the benefits of your graph-based deep learning method, requests more detailed descriptions regarding the construction of the WGCN, the specifics of the real-cell scRNA-seq datasets used, and the extent to which DeepRIG offers new predictions. They also suggest discussing the limitations and potential caveats of your proposed approach and addressing the use of abbreviations in the manuscript. Reviewer 3 appreciates your work as an advance in GRN reconstruction and genomic research. Nevertheless, they suggest discussing works that employ similar data/models to infer GRNs and provide evidence for the transferability of your method to other cell types, conditions, or datasets. Furthermore, they require clarity on the number of interactions used for training and prediction, and if any of the newly predicted interactions are validated by other experiments. Lastly, they recommend a fair comparison of your model with other methods and an explanation of the metrics used in Figure 8B.

We encourage you to consider these comments carefully and revise your manuscript accordingly. We believe these points will greatly improve the quality and presentation of your work.

Reviewer **Comments to the Authors:**

Reviewer #1: The manuscript entitled "Inferring gene regulatory network from single-cell transcriptomes with graph autoencoder model" introduces a graph-based deep learning model called DeepRIG, which is employed on single-cell RNA-sequencing data to infer cell type-specific GRNs. This approach offers the advantage of capturing local and global regulatory structures underlying gene expression, addressing a limitation often overlooked by other GRN inference methods. The performance of DeepRIG is extensively assessed against prior practices for GRNs inference on both in silico and real-cell regulatory networks, showing the benefit of the graph-based deep learning method. In particular, DeepRIG presents accurate cell type-specific GRNs inference for PBMCs and the identification of novel regulators for breast cancer.

I have a few questions:

1. When constructing a WGCN by measuring Spearman's correlation coefficients of each gene pair, the authors defined the edge weight with the absolute value of correlation coefficients. Why did the author adopt this definition strategy rather than other strategies, such as different thresholds? That's what I'm very interested in.

2. While the authors have summarized the six real-cell scRNA-seq datasets, their details and the corresponding ground truth of TF-target interactions still need to be well described, such as the number of TF and target genes in each dataset and the number of links in the corresponding ground truth.

3. It needs to be clarified how much of the results in the PBMC case study are new predictions from DeepRIG and how much is already present in the hTFtarget interactions used for training.

4. The authors have introduced several advantages of their proposed approach. Yet, to facilitate its usage, the authors may also discuss its limitations and potential caveats.

5. The authors need to mention any previous work on identifying regulons from scRNA-seq data, while it is also a practical approach to infer GRNs indirectly.

6. Some abbreviations in the manuscript do not have their complete spelling when they first appear, such as "scRNA-seq data" in Line 16 in the section of Abstract.

Reviewer #2: The comments are in the attached word document.

Reviewer #3: The study introduces DeepRIG, a graph-based deep learning model for inferring gene regulatory networks (GRNs) from single-cell RNA sequencing data. Unlike traditional methods that only consider pairwise gene regulatory relationships, DeepRIG captures the global regulatory structure by transforming gene expression data into a co-expression mode to build a prior regulatory graph. It then employs a graph autoencoder model to embed the global regulatory information into gene latent embeddings and reconstruct the GRN. Benchmarking results show that DeepRIG outperforms existing methods in accurately reconstructing GRNs on completely synthetic and simulated regulatory networks. The model was applied to human peripheral blood mononuclear cells and triple-negative breast cancer samples.

While the manuscript introduces a novel method to improve GRN inference, it lacks clarity on how the WGCNA network and the final GRN differ (and whether those differences are biologically meaningful at all). There are also concerns in the way the comparison to other methods is presented, as discussed below.

Major comments:

1. There is no information on how the WGCNA network and the final GRN differ, especially in the real data presented in figures 7 and 8, where most of the networks have a single central node. The authors should presented clear data on how the GRN differs between the “prior regulatory graph” defined by the WGCNA and the final inferred GRN. Please present what nodes and edges change in figure 7 and 8 if one were to reconstruct the network just using the weights of WGCNA.

2. All the real data comparisons in Figure 5B are unfair because DeepRIG is a supervised model trained on ground truth ChIP-seq data and then evaluated using ChIP-seq data while most of the other methods are unsupervised and do not use ChIP-seq information. This should be clearly stated in the figure and in the main text for this section. Indeed, CNNC, the other supervised method has really similar performance, bringing the question of whether DeepRIG poses a real practical improvement other than being a supervised method.

3. Although the authors explain in lines 565-569 that they only considered undirected regulatory networks because the ChIP-seq data they use does not specify activation or inhibition, single-cell gene expression networks do indirectly contain that information. No directionality nor inhibitory relationships are considered in the current model. In fact, many of the nodes in the presented networks from Figure 3 have been shown to have mutually inhibitory relationships, such as Gata1 and Pu1 in the HSC network. The authors could improve the potential of their manuscript by commenting/expanding their conclusions to positive and negative regulation, which is widespread in real biological networks.

4. Line 154: “to better mimic the zero-flame”. Are the authors referring to “Zero-inflation”? It would be most relevant to model the number of UMIs detected per single-cell too. The number of UMIs (or number of genes, which is positively correlated with the former) is, together with the number of single-cells, an important driver of the sparsity in single-cell datasets.

5. Figure 4. Please change the legend from “no. of cells” to “Dropout rate”.

6. The numbers of cells in figure 5C is very low for current datasets. The authors should benchmark or show how their method scales up this with tens of thousands or hundreds of thousands of cells, which are now routine in single-cell experiments. Similarly, the number of genes detected in an experiment can reach 8000 or more.

7. The authors should also include the results from the other network inference methods in figure 7, and comment on how they differ.

8. Why are only two TFs discussed in figure 7? How about the other TFs expressed in this dataset?

Minor comments

-line 297: what do the authors mean by “top-four cell numbers”?

-Please check inconsistent grammar throughout the whole manuscript. A couple of examples: “PPCOR has the best computational complexity and only spent only a few seconds to complete the GRN “, or “Due to the sizes of scRNA-seq datasets vary from hundreds to millions, ".

**Have all data underlying the figures and results presented in the manuscript been provided?**

Reviewer #1: Yes

Reviewer #2: Yes

Reviewer #3: Yes

PLOS authors have the option to publish the peer review history of their article (what does this mean?). If published, this will include your full peer review and any attached files.

Reviewer #1: No

Reviewer #2: No

Reviewer #3: No

---

## [Decision Letter · Decision Letter 1]

29 Aug 2023

Dear Dr Zou,

We are pleased to inform you that your manuscript entitled "Inferring gene regulatory network from single-cell transcriptomes with graph autoencoder model" has been editorially accepted for publication in PLOS Genetics. Congratulations!

Yours sincerely,

David Balding

Section Editor

PLOS Genetics

Comments from the reviewers (if applicable):

Reviewer's **Comments to the Authors:**

Reviewer #1: The author has addressed all my concerns.

Reviewer #3: My original comments have been addressed.

**Have all data underlying the figures and results presented in the manuscript been provided?**

Reviewer #1: Yes

Reviewer #3: None

PLOS authors have the option to publish the peer review history of their article (what does this mean?). If published, this will include your full peer review and any attached files.

Reviewer #1: No

Reviewer #3: No

**Data Deposition**

http://datadryad.org/submit?journalID=pgenetics&manu=PGENETICS-D-23-00422R1

**Press Queries**

---

## [Editor Report · Acceptance letter]

8 Sep 2023

PGENETICS-D-23-00422R1 

Inferring gene regulatory network from single-cell transcriptomes with graph autoencoder model 

Dear Dr Zou, 

We are pleased to inform you that your manuscript entitled "Inferring gene regulatory network from single-cell transcriptomes with graph autoencoder model" has been formally accepted for publication in PLOS Genetics! Your manuscript is now with our production department and you will be notified of the publication date in due course.

With kind regards,

Anita Estes

PLOS Genetics

On behalf of:
